# Origin of Salt Effects in S_N_2 Fluorination Using KF Promoted by Ionic Liquids: Quantum Chemical Analysis

**DOI:** 10.3390/molecules26195738

**Published:** 2021-09-22

**Authors:** Young-Ho Oh, Sungyul Lee

**Affiliations:** Department of Applied Chemistry, Kyung Hee University, Deogyeong-daero 1732, Yongin City 446-701, Korea; chem_yhoh@daum.net

**Keywords:** S_N_2 fluorination, salt effect, ionic liquid, mechanism

## Abstract

Quantum chemical analysis is presented, motivated by Grée and co-workers’ observation of salt effects [Adv. Synth. Catal. 2006, 348, 1149–1153] for S_N_2 fluorination of KF in ionic liquids (ILs). We examine the relative promoting capacity of KF in [bmim]PF_6_ vs. [bmim]Cl by comparing the activation barriers of the reaction in the two ILs. We also elucidate the origin of the experimentally observed additional rate acceleration in IL [bmim]PF_6_ achieved by adding KPF_6_. We find that the anion PF_6_^−^ in the added salt acts as an extra Lewis base binding to the counter-cation K^+^ to alleviate the strong Coulomb attractive force on the nucleophile F^−^, decreasing the Gibbs free energy of activation as compared with that in its absence, which is in good agreement with experimental observations of rate enhancement. We also predict that using 2 eq. KF together with an eq. KPF_6_ would further activate S_N_2 fluorination

## 1. Introduction

Besides the numerous advantages as solvents, ionic liquids (ILs) [1,2,3,4,5,6,7,8,9] are being used as extremely versatile and efficient organocatalysts/promoters in many chemical reactions [10,11,12,13,14,15,16,17,18,19,20]. This role of ILs for accelerating chemical reaction rates is due to the ionic nature of IL cations and anions, which exert strong electrostatic (Coulombic) forces on the participants in the reaction system. This seems to be especially notable for S_N_2 reactions [10,11,20,21,22,23,24,25,26,27], because in this fundamental process the nucleophile may possess negative charges (halides, CN^−^, NH_2_^−^, OH^−^ etc.) that strongly interact with the relation counter-cations (alkali metal cations, tetraalkyl ammonium cations, etc.) and substrates. IL anions and cations may also interact with the substrates via hydrogen bonding. Elucidating the mechanism of ILs for enhancing the rates and yields of S_N_2 reactions would certainly help to design task-specific ILs by monitoring these interactions. As for S_N_2 fluorination, many schemes have been employed using ILs comprising the 1-butyl-3-methylimidazolium (bmim^+^) or derivatives as the IL cation and IL anion such as OMs^−^, Br^−^, OTf^−^ or PF_6_^−^ and the fluorinating agents such as CsF or KF. This class of ILs are called “phase transfer catalysts” because it looks as if the nucleophile F^−^ takes the place of the IL anion for efficient S_N_2 fluorination. The mechanism of this scheme was elucidated previously by Lee and co-workers [10,22,23,24,25,26,27]. It was observed in numerous studies that the capacity of ILs as catalysts/promoters for S_N_2 reactions depend strongly on the structure of Ils. Kim and co-workers [11,26], for example, found that S_N_2 fluorination may be promoted or suppressed depending on the structure of IL cation: While S_N_2 fluorination may proceed well (yield = 90% in 2 h) in [bmim]PF_6_, the reaction is completely suppressed in [hexaethyleneglycol-mim]PF_6_. They also observed that [bmim]PF_6_ is a much better catalyst than [bmim]OMs or [bmim]OTs, demonstrating the conspicuous influence of the IL anion. 

Grée and co-workers’ experimental study of S_N_2 fluorination by KF in ILs is of high interest in this field. They observed [28] that addition of salts such as KPF_6_ produced notable salt effects in S_N_2 fluorination promoted by imidazolium ionic liquids using KF as the source of F^−^. The origin of these interesting observations have not been fully discussed yet. Their interesting observations are: First, the ILs [bmim]Cl and [bmim]PF_6_ strongly promote S_N_2 fluorination compared with the reaction in organic solvent, in the order of [bmim]PF_6_ > [bmim]Cl. Second, they observed that adding the salt KPF_6_ to the IL reaction medium enhances the promoting capacity of the imidazolium ILs.

Here we present quantum chemical calculations for model systems to theoretically analyze Grée and co-workers’ experiments. We provide a possible explanation for the origin of the experimentally observed additional rate enhancement in ionic liquid [bmim]PF_6_ achieved by adding the salt KPF_6_. We find that the anion PF_6_^−^ in the added salt acts as an extra Lewis base binding the counter-cation K^+^ to the nucleophile, mitigating its strong Coulomb attractive force on F^−^. The effects of the added salt are revealed as decreased Gibbs free energy of activation as compared with that in the absence of salt effects (using 1 eq. of KF). We also show that using 2 eq. of the reactant KF produces effects similar to adding an eq. of KPF_6_. The fluoride in the extra KF acts not as a nucleophile but as an additional Lewis base binding to the counter-cation K^+^ to activate the reaction. 

## 2. Results

Figure 1 and Figure 1 present S_N_2 fluorination experimentally observed by Grée and coworkers. Of the ILs used, [bmim]PF_6_ seems to give better performance as the promoter of the reaction, resulting in a reaction yield of ~86% in 5 h. The corresponding yield in [bmim]Cl was only ~30%, indicating that the latter IL is much less efficient. Figure 1 also shows the activation of fluorination process by adding the salt KPF_6_, by which the reaction essentially completes in 2 h. 

### 2.1. S_N_2 Fluorination without Salt Effects: Using 1 Eq. of KF in [bmim]PF_6_

First, we study the case of using a substrate:KF ratio of 1:1, in which the metal salt acts only as fluorinating agent, that is, as the source of F^−^, thus no salt effects may be ascribed to this situation. Figure 2 presents the transition sates (TSs) and energetics for this process (for coordinates and structures of pre- and post-reaction complexes, see Appendix A. The reference for the zero of the free energy is taken as the ‘free’ reactants, for which the substrate, the CIPs KF, [bmim]PF_6_ or [bmim]Cl are all separated from one another in solution phase). The IL [bmim]PF_6_ and KF are used as the promoter and fluorinating agent, respectively, in the same amount as the substrate (substrate:KF:[bmim]PF_6_ = 1:1:1). We found two reaction routes, case 1 and case 2 with Gibbs free energy of activation *G*^‡^ of 22.8 and 21.4 kcal/mol, respectively. The energetics of the reaction depicted in Figure 2 predicts that the reaction would proceed by the mechanism with lower *G*^‡^ (case 2) according to the Curtin–Henderson principle (When no equilibrium occurs between the free reactants and pre-reaction complex, the reaction proceeds preferably via the path with the lowest Gibbs free energy TS, irrespective of the Gibbs free energies of pre-reaction complexes) [29]. Here, the metal salt KF reacts as a contact ion pair, and electronegative F atoms in the IL anion PF_6_^−^ act as Lewis base coordinating to the counter-cation K^+^, alleviating the latter’s adverse Coulombic influence on the nucleophile F^−^. The main difference between the two mechanisms is the position of the counter-cation K^+^. It seems that in (Case 2), K^+^ is further stabilized by interacting with the electron abundant phenyl ring.

Figure 3 depicts the TSs and the energetics of S_N_2 fluorination under the promoting effects of [bmim]Cl. Again, with a substrate:KF ratio of 1:1. We obtained two alternative routes, of which the (case 2) is more favorable with lower G^‡^ (23.9 kcal/mol). The higher *G*^‡^ than that (21.4 kcal/mol) for the S_N_2 fluorination in [bmim]PF_6_ seems to be in agreement with the experimentally observed lower reaction yield (~30%) as compared with that (~90%) in [bmim]PF_6_. This difference in the promoting influence of the two ILs may result from the difference in the ability of the IL anions Cl^−^ and PF_6_^−^: More electronegative and numerous F atoms in PF_6_^−^ may donate partial negative charges to the counter-cation K^+^ much better than Cl^−^.

### 2.2. Salt Effects: Adding KPF_6_ or Using 2 Eq. of KF 

The most proper question to ask would be: What is the mechanism of S_N_2 rate enhancement by the added salt by KPF_6_? Figure 4 presents the TSs and energetics for S_N_2 fluorination promoted by [bmim]PF_6_ and activated by KPF_6_. The role of the added salt may be seen clearly from the structures of the TSs: In both TSs the anion PF_6_^−^ binds to the two K^+^, acting as an additional Lewis base on the counter-cation K^+^ to the nucleophile F^−^. As a result, the Gibbs free energy of activation now decreases to 16.4 kcal/mol from that (21.4 kcal/mol) in the absence of salt effects given in Figure 2. We think that this is the origin of rate enhancement by adding KPF_6_ observed by Grée and co-workers. In the presence of additional KPF_6_, more electrostatic interactions are allowed (PF_6_^−^…K^+^…F^−^, PF_6_^−^ …bmim^+^ ring, and PF_6_^−^…K^+^… PF_6_^−^) than in its absence, thus stabilizing the TS. For example, the natural bond orbital charge of the H atom nearest to PF_6_^−^ decreases from +0.283 (Figure 2a) to +0.264 (Figure 2a), clearly showing the electrostatic influence of the anion.

If this is the case, then it can be expected that any salt, including KF, may also do, as long as its anion is capable of influencing the reaction as a Lewis base. Indeed, Grée and co-workers used 2 eq. of KF in their experiments with excellent S_N_2 yields. In order to examine this case, we carried out calculations for S_N_2 fluorination in [bmim]PF_6_ with a substrate:KF ratio of 1:2. Figure 5 shows the TS for the most feasible reaction pathway and the corresponding energetics for S_N_2 fluorination. The role of additional eq. KF is clearly seen: The extra F^−^ acts not as a nucleophile, but as an extra Lewis base on the counter-cation K^+^ to the nucleophile, with the two K^+^s and two F^−^s forming a rectangular configuration. 

### 2.3. Salt Effects: Using 2 KF plus 1 Eq. KPF_6_ in [bmim]PF_6_

Finally, we examine the most complicated system, in which two eq. KF and 1 eq. KPF_6_ are used for S_N_2 fluorination in [bmim]PF_6_. In this situation, KF and KPF_6_, each of 1 eq., may activate the reaction in collaboration. Figure 6 describes the TS for the most favorable mechanism and the corresponding energetics. The two anions F^−^ and PF_6_^−^ now help to reduce the Coulombic influence of the counter-cation K^+^, further lowering the *G*^‡^ to 9.1 kcal/mol.

## 3. Computational Details

The M06-2X/6-311G** method [30,31,32] was employed as implemented in Gaussian16 [33]. We adopted the cluster/continuum approximation [8] (accounting for the full solvent effect [34] would require a molecular dynamics approach incorporating a few hundred explicit solvent molecules under periodic boundary condition), including the effects of the solvent continuum by the SMD-PCM method [35]. For the values of the dielectric constants of [bmim]Cl and [bmim]PF_6_, we used 15.0 and 14.0, respectively [36]. We carried out an extensive search for stationary states over the potential energy surface of the system (substrate plus 1 or 2 IL unit). Pre-reaction and post-reaction complexes were obtained by verifying that all harmonic frequencies be real. Transition states were obtained by ascertaining the imaginary frequency of the reaction coordinate, and also by performing the intrinsic reaction coordinate analysis.

## 4. Conclusions

We presented a quantum chemical analysis to account for the activation of S_N_2 fluorination by added salts. The role of the anion (F^−^ or PF_6_^−^) of the added salt KF or KPF_6_^−^ seems to be an extra Lewis base acting on the counter-cation K^+^ to enhance the rate constants. These features of fluorination are in line with our proposed S_N_2 mechanism in which the metal salt reacts as a contact ion-pair [37,38,39,40] and the counter-cation (alkali metal cation) is ‘neutralized” by the Lewis base promoter. 

## Data Availability

Not applicable.

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
