# Peer review of "Origin of Salt Effects in SN2 Fluorination Using KF Promoted by Ionic Liquids: Quantum Chemical Analysis"

_molecules, 2021, doi:10.3390/molecules26195738_

Round 1

Reviewer 1 Report

The manuscript “Origin of Salt Effects in SN2 Fluorination using KF Promoted by Ionic Liquids: Quantum Chemical Analysis” by Young-Ho Oh and Sungyul Lee reports a computational study aimed at interpreting a series of experimental results concerning the fluorination promoted by KF in Ionic Liquids (IL) and in the presence of the salt KPF6 which both increase the efficiency of the reaction if compared to the results in conventional organic solvents.

The manuscript might be of interest for “Molecules” readership, it is fairly clear and the conclusions seem coherent with the presented results. Moreover the level of the calculations appear as adequate for the problem at hand. At the same time, before the manuscript can be accepted, there is a number of minor issues that the authors should address/clarify:

1) In all the presented cases we systematically observe the presence of a formed contact ion pair between K+ and F-. I’m wondering how large is the thermodynamic stability of such a contact ion-pair for a salt dissolved at a relatively low concentration in IL. In conventional, i.e. molecular, liquids the formation of a these species would require a relatively large formation free-energy. Have the authors some evidences supporting the presence of such a species in the experimental conditions? Otherwise the free energy values reported in the Figures should dramtically shift toward more positive values.

2) Related to the previous point: the authors did not specify what is their reference state for the zero of the free energy diagrams. This is a very important aspect because an incosistent choice of the reference state, although probably irrelevant for the relative efficiencies of the presented reaction routes, might completely alter the absolute free energy barriers which should be consistent with the experimental rate constants (or at least with the experimental reaction times). In this respect I would suggest, if the authors didn’t do it, to always use the 1.0 mole/liter as standard, i.e. reference, condition for properly take into account the relative roto-translational entropic terms which, in this case, is expected to be somewhat important.

3) It is not usual to report, in the computational details, that “we have verifyed that all infrared frequencies are real”. As a matter of fact what the authors have done is to calculate the HARMONIC frequencies and not the INFRARED frequencies (which is the subpart of the former with non-zero transition dipole).

4) The literature concerning the use of mean-field (PCM and similar stuff) in the modelling of chemical reactions in IL is rather insufficient (see for example the extensive work from the group of the late Prof. C. Chiappe from the University of Pisa and the huge amount of cited literature).

Author Response

We thank the Reviewer for many helpful comments.

Reviewer 1

  1. In all the presented cases we systematically observe the presence of a formed contact ion pair between K+ and F-. I’m wondering how large is the thermodynamic stability of such a contact ion-pair for a salt dissolved at a relatively low concentration in IL. In conventional, i.e. molecular, liquids the formation of these species would require a relatively large formation free-energy. Have the authors some evidences supporting the presence of such a species in the experimental conditions? Otherwise the free energy values reported in the Figures should dramatically shift toward more positive values.

This is a great question. Actually, none has observed the contact ion-pair SN2 mechanism in

unambiguous terms yet. Only indirect and fragmentary evidences exist, for example, the XRD

structural determination of complex of KF and a 1,1′-bi-2-naphthol(BINOL)-based chiral analog of

oligoethylene glycol presented in Ref. 24, and promotion of SN2 fluorination by 1-Butyl-3-

methylimidazolium fluoride ([Bmim]F) in solvent-free conditions reported by Magnier and co-workers

[Tetrahedron Lett. 55 (2014) 826] (which we add as Ref. 39). Intensive experimental works (Ref. 40)

by the Sessler’s group are also notable, but their studies are for the F-receptor, not for SN2 reactions.

Theoretical predictions were made by Streitwieser (Ref. 37), Bickelhaupt, (Ref. 38) and our lab.

Experimental observation for the CIP SN2 mechanism is our grand plan, and we are progressing toward

it.

  1. Related to the previous point: the authors did not specify what is their reference state for the zero of the free energy diagrams. This is a very important aspect because an inconsistent choice of the reference state, although probably irrelevant for the relative efficiencies of the presented reaction routes, might completely alter the absolute free energy barriers which should be consistent with the experimental rate constants (or at least with the experimental reaction times). In this respect I would suggest, if the authors didn’t do it, to always use the 1.0 mole/liter as standard, i.e. reference, condition for properly taking into account the relative roto-translational entropic terms which, in this case, is expected to be somewhat important.

The reference state for the zero of the free energy is taken in our work as the ‘free’ reactants: The

substrate, the CIP KF, [bmim]PF6 and [bmim]Cl, all separated from one another in solution phase. We

make this explicit by adding a paragraph, pp. 2, pp. 3, line #5-7:

The reference for the zero of the free energy is taken as the ‘free’ reactants, for which the

substrate, the CIPs KF, [bmim]PF6 or [bmim]Cl are all separated from one another in solution phase.

  1. It is not usual to report, in the computational details, that “we have verified that all infrared frequencies are real”. As a matter of fact what the authors have done is to calculate the HARMONIC frequencies and not the INFRARED frequencies (which is the subpart of the former with non-zero transition dipole).

We revise the sentence by correcting the ‘infrared frequencies’ to ‘harmonic frequencies’

  1. The literature concerning the use of mean-field (PCM and similar stuff) in the modelling of chemical reactions in IL is rather insufficient (see for example the extensive work from the group of the late Prof. C. Chiappe from the University of Pisa and the huge amount of cited literature).

Although we are well aware of the limitations of the PCM method, it is the best we could do. Full treatment of the solvent effects would involve the molecular dynamics calculations including the explicit solvent molecules, but this is beyond of the scope of our work. We add the following paragraph, in Computationad details, pp. 6, line # 20 –pp. 7, line# 2, citing Chiappe and co-workers’ work.:

accounting for the full solvent effect [34] would require molecular dynamics approach incorporating a few hundred explicit solvent molecules under periodic boundary condition.

Reviewer 2 Report

In this manuscript, the authors theoretically examined the fluorination reaction by KF in ionic liquids following a recent experimental finding on it. As a result, they succeeded to obtain the reaction mechanisms supporting the experimental result. In the manuscript, the calculations are appropriately performed and the conclusion is reasonable. So, I recommend that this manuscript becomes publishable. However, it requires several revisions to be published as follows:

1. The authors should discuss the roles of KPF6 in more detail. The calculated reaction profiles indicate that the presence of KPF6 significantly contributes to the stabilization of the pre-reaction complexes compared to the reactants. Based on the results, the authors should explain the cause for the stabilization by molecular orbital or population analyses.

2. In the presence of KPF6, one K+ ion bonding F- is dissociated from the F- ion of KF and then it attaches the ring of the bmim. I guess this attachment leads to the stabilization of the pre-reaction complex. The authors should explain why K+ ion attaches the ring only in the presence of KPF6.

3. The Curtin-Henderson principle is not well-known for many readers.

4. The "bmim" should be written as 1-butyl-3-methylimidazolium at the first appearance.

5. A few typos are included.

Author Response

We thank the Reviewer for many helpful comments.

Reviewer 2

  1. The authors should discuss the roles of KPF6 in more detail. The calculated reaction profiles indicate that the presence of KPF6 significantly contributes to the stabilization of the pre-reaction complexes compared to the reactants. Based on the results, the authors should explain the cause for the stabilization by molecular orbital or population analyses.

In Figure 4 (a) (Case 2), it is obvious that there exist more electrostatic interactions in the presence of additional KPF6 than in its absence. We carried out Natural Bond Orbital analysis for the partial charge of the H atom interacting with the PF6- anion. We also changed the orientation of the Figure, and add the following paragraph, pp. 5, line # 6-10.

In the presence of additional KPF6, more electrostatic interactions are allowed (PF6-…K+…F-, PF6-- …bmim+ ring, and PF6-…K+… PF6-) than in its absence, thus stabilizing the TS. For example, the natural bond orbital charge of the H atom nearest to PF6- decreases from +0.283 (Figure 2 (a)) to +0.264 (Figure 2 (a)), clearly showing the electrostatic influence of the anion.

  1. In the presence of KPF6, one K+ ion bonding F- is dissociated from the F- ion of KF and then it attaches to the ring of the bmim. I guess this attachment leads to the stabilization of the pre-reaction complex. The authors should explain why K+ ion attaches the ring only in the presence of KPF6.

In Figure 4 (a) (Case 1), it looks as if the K+ binds to the bmim+ ring, but it does not. We changed the orientation of the structure in the Figure to make this clear.

  1. The Curtin-Henderson principle is not well-known for many readers.

We add the following paragraph explaining the Curtin-Henderson principle, pp. 3, line # 12-14:

Curtin-Henderson Principle (When no equilibrium occurs between the free reactants and pre-reaction complex, the reaction proceeds preferably via the path with the lowest Gibbs free energy TS irrespective of the Gibbs free energies of pre-reaction complexes)

  1. The "bmim" should be written as 1-butyl-3-methylimidazolium at the first appearance.

We add the full name of "bmim", pp. 1, line # 32-33: 1-butyl-3-methylimidazolium (bmim+)

  1. A few typos are included.

Typos are corrected:

Pp. 2, line # 29, sulfolane is deleted.

Pp. 5, line # 5, 20.1 à 21.4 kcal/mol
